# Feasibility of Perineal Defect Reconstruction with Simplified Fasciocutaneous Inferior Gluteal Artery Perforator (IGAP) Flaps after Tumor Resection of the Lower Rectum: Incidence and Outcome in an Interdisciplinary Approach

**DOI:** 10.3390/cancers15133345

**Published:** 2023-06-26

**Authors:** J. T. Thiel, H. L. Welskopf, C. Yurttas, F. Farzaliyev, A. Daigeler, R. Bachmann

**Affiliations:** 1Department of Hand, Plastic, Reconstructive and Burn Surgery, BG Unfallklinik Tuebingen, University of Tuebingen, 72076 Tuebingen, Germany; 2General, Visceral and Transplantation Surgery, Eberhard-Karl-Universität, University Hospital Tübingen, 72076 Tuebingen, Germany

**Keywords:** perineal VY advancement flap, perineal wound closure, perineal wound complications, rectal cancer, inferior gluteal artery perforator flaps

## Abstract

**Simple Summary:**

Surgery of locally advanced rectal cancer involving the anal sphinkter or recurrent anal cancer demands reliable surgical margins resulting in local defects. Skin-soft tissue reconstruction after extended or extralevator approaches to abdominoperineal resection for lower gastrointestinal tract cancers or inflammatory tumors has been described mainly with muscle flaps such as the vertical rectus abdominis muscle flap. In this article, we demonstrate our tailored approach with the use of bilateral adipo-fasciocutaneous perforator flaps of the inferior gluteal artery in a VY manner after surgery in 29 cases. The relatively simple surgical technique and postoperative course are presented. Surgery was performed with a minimally invasive abdominal approach combined with open perineal extralevator abdominoperineal resection and immediate flap reconstruction by a plastic surgeon. The rate of early perineal complication after plastic reconstruction was 19.0%, requiring local revision due to local infection. The use of bilateral adipo-fasciocutaneous perforator flaps of the inferior gluteal artery is a reliable, rapid, and safe option for pelvic floor reconstruction with minimal donor site morbidity.

**Abstract:**

Background: Extralevator abdominoperineal excision (ELAPE) is a relatively new surgical technique for low rectal cancers, enabling a more radical approach than conventional abdominoperineal excision (APE) with a potentially better oncological outcome. To date, no standard exists for reconstruction after extended or extralevator approaches of abdominoperineal (ELAPE) resection for lower gastrointestinal cancer or inflammatory tumors. In the recent literature, techniques with myocutaneous flaps, such as the VY gluteal flap, the pedicled gracilis flap, or the pedicled rectus abdominis flaps (VRAM) are primarily described. We propose a tailored concept with the use of bilateral adipo-fasciocutaneous inferior gluteal artery perforator (IGAP) advancement flaps in VY fashion after ELAPE surgery procedures. This retrospective cohort study analyzes the feasibility of this concept and is, to our knowledge, one of the largest published series of IGAP flaps in the context of primary closure after ELAPE procedures. Methods: In a retrospective cohort analysis, we evaluated all the consecutive patients with rectal resections from Jan 2017 to Sep 2021. All the patients with abdominoperineal resection were included in the study evaluation. The primary endpoint of the study was the proportion of plastic reconstruction and inpatient discharge. Results: Out of a total of 560 patients with rectal resections, 101 consecutive patients with ELAPE met the inclusion criteria and were included in the study evaluation. The primary direct defect closure was performed in 72 patients (71.3%). In 29 patients (28.7%), the defect was closed with primary unilateral or bilateral IGAP flaps in VY fashion. The patients’ mean age was 59.4 years with a range of 25–85 years. In 84 patients, the indication of the operation was lower rectal cancer or anal cancer recurrence, and non-oncological resections were performed in 17 patients. Surgery was performed in a minimally invasive abdominal approach in combination with open perineal extralevatoric abdominoperineal resection (ELAPE) and immediate IGAP flap reconstruction. The rate of perineal early complications after plastic reconstruction was 19.0%, which needed local revision due to local infection. All these interventions were conducted under general anesthesia (Clavien–Dindo IIIb). The mean length of the hospital stay was 14.4 days after ELAPE, ranging from 3 to 53 days. Conclusions: Since radical resection with a broad margin is the standard choice in primary, sphincter-infiltrating rectal cancer and recurrent anal cancer surgery in combination with ELAPE, the choice technique for pelvic floor reconstruction is under debate and there is no consensus. Using IGAP flaps is a reliable, technical, easy, and safe option, especially in wider defects on the pelvic floor with minimal donor site morbidity and an acceptable complication (no flap necrosis) rate. The data for hernia incidence in the long term are not known.

## 1. Introduction

Locally advanced low rectal and recurrent anal carcinomas are the most common indications for extralevatoric rectal excision. In the last decade, oncological rectal cancer treatment has changed enormously, especially in the era of multimodal treatment using radiochemotherapy and chemotherapy in total neoadjuvant therapy [1]. Total mesorectal excisions (TMEs), chemo radiation, and recently, more radical surgical techniques such as extralevator abdominoperineal excisions (ELAPEs) have been introduced to address these issues and to improve oncological outcomes in low rectal cancers while avoiding tumor perforation and positive circumferential margins [2,3,4]. Chronic inflammatory conditions are additional reasons for abdominoperineal excision. In the case of cancer, a combination with radiochemotherapy is common in neoadjuvant or initial definitive treatment. The techniques of radical rectal resection have changed by addressing local poor oncological outcomes regarding disappointing results of conventional abdominoperineal excision (APE) with respect to positive circumferential resection margins (CRMs) and local recurrence rates [5]. Beside oncological surgery, in benign disease, perianal fistulating Crohn disease is one possible cause of extralevatoric excision. The aim of the procedure is to move the area of resection as far as possible from the anorectal junction, which is the area at highest risk of perforation and CRM positivity. Meanwhile, ELAPE involves a total mesorectal excision up to the coccyx and the pelvic peritoneal dissection anterior to the Denonvilliers’ fascia. After colostomy, the abdomen is closed. In the modified prone jack-knife position, the gluteus maximus and levators are then divided laterally. The endopelvic fascia is divided, and the pelvic dissection is continued anterior to the Denonvilliers’ fascia, delivering a cylindrical specimen. Hereby, the anterior region is also addressed with comfortable access to the dissection area by placing the patient in a modified prone jack-knife position. These techniques result in large local defect zones. The resulting perineal defects following resection represent a significant reconstructive challenge. A variety of flaps have been described, mainly using the abdominal, gluteal, and pudendal sites [6,7,8]. Of these, abdominal myocutaneous (VRAM) flaps, are most commonly used and are widely reported in the literature [9,10,11,12,13].

However, technical advancements in modern rectal surgery have minimized abdominal morbidity from both the resection and the reconstruction. Minimally invasive techniques are increasingly adopted in laparoscopic or robotic-assisted surgery. In abdominoperineal resection, the minimally invasive approach in combination with an open local excision and the classical VRAM technique was not the best option due to the abdominal wall morbidity with a need for colostomy/urostomy and the prone jack-knife position of the patient. Numerous alternative techniques to the VRAM flap have been described, predominantly using the abdominal, pudendal, gluteal, and thigh donor sites [6,8,14,15,16,17,18,19]. Among those is the myocutaneous gracilis flap, which is a well-described alternative to the VRAM flap for genital and perineal reconstruction. As an alternative reconstructive strategy addressing the dead space obliteration to prevent the risk of intestinal prolapse with access for patients in a prone jack-knife positioning, perforator-based local flaps of the perineal and gluteal region have been recently introduced in perineal coverage. One of these is the fasciocutaneous gluteal flap, which is based on inferior gluteal artery perforators (IGAPs) [7,14,15]. In addition to the oncological indications, perianal fistulous Crohn disease and chronic inflammatory conditions in benign diseases are also possible causes of extralevatoric excisions using the established surgical pathways. This strategy is almost identical to oncological resection. This paper presents our experience with the uni- or bilateral IGAPs for reconstructing perineal defects after the resection of anorectal malignancies and compares the outcome regarding the clinical key parameters.

## 2. Material and Methods

### 2.1. Patient and Study Design

Based on the German procedure classification (OPS) codes 5-484 and 5-485, a total of 554 consecutive patients were identified from 2017 to September 2021. Among them, 101 consecutive patients with rectal extirpation met the inclusion criteria. The data on successive patients undergoing ELAPEs with or without plastic reconstruction were collected retrospectively at the University of Tuebingen, Germany, during the study period from January 2017 to September 2021. The data collected included that patients’ demographics, pathology, chemoradiotherapy, type and duration of surgery, length of hospital stay, postoperative complications (flap-related and general), and duration of hospital stay. Early complications were defined as wound dehiscence, partial flap necrosis, wound infections, and fluid retention in the wound within the postoperative days until discharge. Ultimately, 101 patients were included in this study, of whom 29 underwent IGAP flap coverage following malignoma or a fistulating inflammatory disease resection of the lower rectum with perineal involvement.

### 2.2. Operative Procedure

After the minimally invasive mobilization of the descending colon, the total mesorectal excision, and the creation of the end colostomy, the abdomen was closed, and the patient was turned to the recumbent jack-knife position for perineal dissection. This was followed by the planning of the flap reconstruction and the plastic defect coverage by a senior plastic surgeon of our department. The perforator flap of the inferior gluteal artery (IGAP) was marked, as shown in Figure 1, depending on the size of the primary defect and the available tissue volume. The flap tip was designed over the greater trochanter. The adipo-fasciocutaneous flap was harvested from the unilateral or bilateral buttock depending on the size of the defect and the available tissue thickness (Figure 2, Figure 3 and Figure 4). Radiogenically preloaded tissue should not be included in the flap whenever possible. Afterwards, the flap elevation was performed from the lateral to the medial region, with the inclusion of the M. gluteus fascia. The flaps could then easily be mobilized several centimeters medially in a subfascial plane without an extensive visualization or dissection of the perforators. The surgical concept could be simplified by eliminating the time-consuming microsurgical dissection of the perforators. Depending on the required plumbing of the wound cavity, the de-epithelialization of the medial flap portion or—in the bilateral approach—that of one medial flap portion was then performed (Figure 2). The insertion of the deepithelialized flap base was finally performed with delayed absorbable suture material—e.g., Vicryl 0—for the deep layer. At this point, it was imperative to ensure that both the ureters and the loop of the bowel had not been sutured into place. We emphasize the need to transpose the deepithelialized flap as deeply as possible into the pelvis and suture it onto the sacrum, presacral ligaments, and remaining pelvic outlet. The donor site was closed in V–Y fashion, and the remaining flap portions were sutured in three layers with absorbable sutures for Scarpa's fascia and the deep dermal layer. Medially, the skin of the flap plastics was sutured using a deeply penetrating, non-absorbable material—e.g., Prolene 0—via the Donati technique (vertical mattress suture) in the area with the greatest tension. Unlike continuous sutures, the above technique allows for individual sutures to be opened in the event of fluid accumulation at depth or wound healing disorders. Laterally, the skin was closed using intracutaneous sutures or skin staples. Previously, Redon drainages were inserted for both the deep wound cavity and the superficial wound areas with bilateral drainage. We left the non-absorbable suture material in place for 21 days, and the redon drains for at least 7 days, with an expected prolonged drainage phase. After surgery, supine positioning was allowed for 5 days, starting with sitting training on day 6. The patients were mobilized on the first postoperative day.

### 2.3. Statistical Analysis

The statistical analysis was conducted using the SPSS software, version 21 (SPSS Inc., Chicago, IL, USA). The results were reported as the medians and lower and upper quartiles. The survival curves were calculated using a Kaplan–Meier analysis and the log rank test. The significant differences between the examined groups, the Student’s *t* test, and the Mann–Whitney U test were used. The significance level was defined as *p* < 0.05.

## 3. Results

### 3.1. Study Group and Demographics (Table 1)

Between 2017 and September 2021, 101 patients with abdominoperineal resection, among whom 71 were male (71.3%) and 29 were female (28.7%), were included in the study. Their mean age was 59.4 years, and their mean BMI was 25.7 kg/m^2^. Eighty-four patients (83.2%) had an ECOG performance score of 0, 15 had a score of 1 (14.9%), one had a score of 2 (1.0%), and one a score of 3 (1.0%). The indicators for rectal extirpation were rectal adenocarcinoma in 63 cases (62.4%), inflammatory bowel disease (Crohn’s disease and ulcerative colitis) in 15 (14.9%), anal squamous cell carcinoma in 11 (10.9%), melanoma in three (3.0%), sarcoma in two (2.0%), gastrointestinal stromal tumors (GIST) in two (2.0%), neuroendocrine neoplasia/carcinoma (NEN/NEC) in two (2.0%), chronic anastomotic leakage after rectal resection in two (2.0%), and low anterior resection syndrome (LARS) after rectal resection in one patient (1.0%). The initial clinical tumor staging in the oncological cases revealed one patient with cT1 (1.4%), nine with cT2 (12.3%), 42 with cT3 (57.5%), and 21 with cT4 (28.8).

**Table 1 cancers-15-03345-t001:** Characteristics of the study sample.

Characteristic	*n* (%)
Mean age (range)	59.4 (25–85)
Gender	
Female	29 (28.7)
Male	72 (71.3)
Mean BMI (range)	25.7 (15.2–41.8)
ECOG	
0	84 (83.2)
1	15 (14.9)
2	1 (1.0)
3	1 (1.0)
Indication	
Rectal adenocarcinoma	63 (62.4)
Inflammatory bowel disease (IBD)	15 (14.9)
Anal squamous cell carcinoma	11 (10.9)
Melanoma	3 (3.0)
Sarcoma	2 (2.0)
GIST	2 (2.0)
NEN/NEC	2 (2.0)
Chronic anastomotic leakage	2 (2.0)
LARS	1 (1.0)
cT category	
cT1	1 (1.4)
cT2	9 (12.3)
cT3	42 (57.5)
cT4	21 (28.8)
cN category	
cN0	28 (37.3)
cN1	47 (62.7)
cM category	
cM0	64 (77.1)
cM1	19 (22.9)

### 3.2. Treatment Parameters (Table 2)

Abdominoperineal rectal excision was performed on 72 patients, and pelvic exenteration was performed on the remaining 29. The indicators for abdominoperineal rectal excision and the terminal stoma creation were rectal adenocarcinoma in 47 patients (65.3%), anal squamous cell carcinoma in nine, inflammatory bowel disease (IBD, Crohn’s disease, ulcerative colitis) in eight (11.1%), melanoma in two (2.7%), chronic anastomotic leakage after rectal resection in two (2.7%), GIST in one (1.4%), NEN/NEC in one (1.4%), sarcoma in one (1.4%), and LARS in one patient (1.4%). The indicators for exenteration surgery—including abdominoperineal rectal excision and the creation of the terminal stoma—were rectal adenocarcinoma in 16 (55.2%), IBD in seven (24.1%), anal squamous cell carcinoma in two (6.9%), GIST in one (3.4%), NEN/NEC in one (3.4%), sarcoma in one (3.4%), and melanoma in one patient (3.4%). Regarding the surgical access, abdominoperineal excisions were performed laparoscopically in 50 patients (69.4%) and robotically assisted in 12 patients (16.7%). In total, 62 minimally invasive surgical accesses (86.1%) were enabled. Five patients with rectal extirpation had open surgical access (5.9%) and five had perineal access (5.9%). For the pelvic exenterations, the surgical access was open in 16 patients (76.2%), laparoscopically assisted in eight (13.8%), converted to open in three (10.3%), robotically assisted in one (3.4%), and perineal in one patient (3.4%). Either plastic reconstruction or direct suturing was performed. In 29 patients (28.7%), local defects had to be covered by IGAP flaps, while in the remaining 72 (71.3%), perineal defects were closed using direct suturing of the subcutaneous tissue. All the surgeries included the creation of a terminal stoma—a terminal ileostomy in 11 patients (10.9%) and a terminal descendostomy in the other 90 (89.1%). Treatment-wise, 66 of the patients were treated with neoadjuvant therapy, 39 with total neoadjuvant therapy (47.0%), 14 with radiochemotherapy (16.9%), 10 with short-term radiotherapy (12.0%), two with Imatinib (1.4%), and one with chemotherapy (1.2%). A further 17 patients were treated without neoadjuvant therapy (20.5%). Moreover, 25 oncological patients were treated with chemotherapy (30.1%) as adjuvant therapy, five with radiochemotherapy (6.0%), two with immunotherapy (2.4%), and one with Imatinib (1.2%). For two of the patients, it was unknown whether they received adjuvant therapy (they did not receive adjuvant therapy at UKT, and no subsequent data were available after their surgery and discharge). Forty-eight patients (57.8%) did not receive any adjuvant therapy.

**Table 2 cancers-15-03345-t002:** Treatments administered.

Treatment	*n* (%)
Abdominoperineal excision	72
Rectal adenocarcinoma	47 (65.3)
Anal squamous cell cancer	9 (12.5)
Inflammatory bowel disease (IBD)	8 (11.1)
Melanoma	2 (2.7)
Chronic anastomotic leakage	2 (2.7)
GIST	1 (1.4)
NEN/NEC	1 (1.4)
Sarcoma	1 (1.4)
LARS	1 (1.4)
Exenteration surgery	29
Rectal adenocarcinoma	16 (55.2)
IBD	7 (24.1)
Anal squamous cell cancer	2 (6.9)
GIST	1 (3.4)
NEN/NEC	1 (3.4)
Sarcoma	1 (3.4)
Melanoma	1 (3.4)
Surgical access	
Laparoscopically assisted	58 (57.4)
Open	21 (20.8)
Robotically assisted	13 (12.9)
Perineal	6 (5.9)
Conversion to open	3 (3.0)
Surgical access and surgical procedure	
Abdominoperineal excision:	72
Laparoscopically assisted	50 (69.4)
Open	5 (6.9)
Robotically assisted	12 (16.7)
Perineal	5 (6.9)
Pelvic exenteration:	29
Laparoscopically assisted	8 (13.8)
Open	16 (76.2)
Robotically assisted	1 (3.4)
Perineal	1 (3.4)
Conversion to open	3 (10.3)
Reconstruction	
VY flap	29 (28.7)
Direct suture	72 (71.3)
Stoma creation	
Ileostomy	11 (10.9)
Descendostomy	90 (89.1)
Neoadjuvant therapy	
Total neoadjuvant therapy	39 (47.0)
Radiochemotherapy	14 (16.9)
Short-term radiotherapy (5xGy)	10 (12.0)
Imatinib	2 (1.4)
Chemotherapy	1 (1.2)
None	17 (20.5)
Adjuvant therapy	
Chemotherapy	25 (30.1)
Radiochemotherapy	5 (6.0)
Immunotherapy	2 (2.4)
Imatinib	1 (1.2)
None	48 (57.8)
Unknown	2 (2.4)

### 3.3. Postoperative Oncological Outcome (Table 3)

The results of the study subjects with abdominoperineal excisions both with and without pelvic exenteration revealed 57 patients without flap reconstruction, 48 (87.3%) with an R0 resection, and seven (12.7%) with an R1 resection. In the group with the VY flap reconstruction, the resection status of 24 patients was documented, among which 23 patients (95.8%) had an R0 resection, while the remaining patient (4.2%) had an R1 resection.

**Table 3 cancers-15-03345-t003:** Oncological outcome of all 101 patients (rectal extirpation and exenteration).

Oncological Outcome	*n* (%)		
pT category			
ypT0	8 (10.1)		
pT1	4 (5.1)		
pT2	20 (25.3)		
pT3	35 (55.3)		
pT4	12 (15.2)		
pN category			
pN0	49 (61.3)		
pN1	21 (26.3)		
pN2	10 (12.5)		
	Direct suture	VY flap	*p* value
Resection status	57	24	0.246
R0	48 (87.3)	23 (95.8)	
R1	7 (12.7)	1 (4.2)	
Circumferential resection margin	43	21	0.957
CRM−	31 (72.1)	15 (71.4)	
CRM+	8 (18.6)	5 (23.8)	
R1	4 (9.3)	1 (4.8)	
Mercury grade (TME quality)	46	20	0.388
Grade 1 (good)	24 (52.2)	8 (40.0)	
Grade 2 (moderate)	13 (28.3)	7 (35.0)	
Grade 3 (poor)	9 (19.6)	5 (25.0)	
Dworak regression grade	29	14	0.088
Grade 1	10 (34.5)	2 (14.3)	
Grade 2	13 (44.8)	6 (42.9)	
Grade 3	4 (13.8)	4 (28.6)	
Grade 4	2 (6.9)	2 (14.3)	

A chi-squared test was used to compare the resection status and the reconstruction of the pelvic floor among the participants. The results suggested a non-significant connection between the resection status and the reconstruction, where *p* = 0.246. In the subgroup of patients with only abdominoperineal excision, the group with the sutured reconstruction showed a resection status of R0 in 38 patients (92.7%), while three patients (7.3%) experienced an R1 resection. In the subgroup with IGAP flap reconstruction, 19 patients (95.0%) had an R0 resection, and one patient (5.0%) experienced an R1 resection. A chi-squared test was used to compare the resection status and the reconstruction of the pelvic floor. The results revealed a non-significant connection between the resection status and the reconstruction, where *p* = 0.731.

### 3.4. Circumferential Margin (CRM), Total Mesorectal Excision (TME), and Dworak Regression

In the group without flap surgery, CRM was applicable in 33 patients, among whom 27 (81.8%) were CRM−, four (12.1%) were CRM+, and two (6.1%) were R1. For the VY flap surgery group, CRM was applicable in 21 patients, of whom 15 (71.4%) were CRM−, five (23.8%) were CRM+, and one (4.8%) was R1. A Mann–Whitney U Test or a Kruskal–Wallis test was used to determine whether there were differences in the CRMs between the direct suture and the VY flap reconstruction. No statistically significant differences were found, where *p* = 0.957. Total mesorectal excisions were applicable to 46 patients in the group without flap surgery, among whom 24 (52.2%) were Mercury grade 1, 13 (28.3%) were grade 2, and nine were grade 3 (19.6%). For the VY flap group, TMEs were applicable to 20 patients, of whom eight (40%) were Mercury grade 1, eight (35%) were grade 2, and five were grade 3 (25%). As before, a Mann–Whitney U Test or a Kruskal–Wallis test was calculated to determine whether there were differences in the TME quality between the primary suturing and the direct flap reconstruction, and no statistically significant difference was found (*p* = 0.388). A Dworak tumor regression grading was applicable to 29 patients in the group without flap surgery, among whom 10 (34.5%) were grade 1, 13 were grade 2 (44.8%), four (13.8%) were grade 3, and two (6.9%) were grade 4. Of the 14 applicable patients in the group with flap surgery, two (14.3%) were grade 1, six were grade 2 (42.9%), four (28.6%) were grade 3, and two (114.3%) were grade 4. The Mann–Whitney U Test or the Kruskal–Wallis test for *the* differences in the Dworak regression grade between the direct suture and the VY flap reconstruction once again revealed no statistically significant difference, where *p* = 0.078.

### 3.5. Operative Surgical Outcome (Table 4)


Duration of abdominoperineal excision


The operative time for an abdominoperineal excision was approximately normally distributed, as assessed by the Shapiro–Wilk test (*p* > 0.05; *p* = 0.283 for the direct sutures, and *p* = 0.932 for the VY flaps). There were 51 patients in the group without flap reconstruction (n = 51) and 21 patients in the group with IGAP flap reconstruction (n = 21). The operative time was shorter in the first group (mean duration = 258.24 min) than in the second (mean duration = 361.52 min). There was a statistically significant difference in the mean operative times of 103.3 min (95% CI = [53.05;153.05]) between the two groups, where *p* < 0.001.

**Table 4 cancers-15-03345-t004:** Surgical outcomes for rectal extirpation.

Surgical Outcome	Direct Suture	VY Flap	*p* Value
Duration	mean 258.24 min	mean 361.52 min	< 0.001
(Range)	(67–575 min)	(222–526 min)	(*t* test)
Length of stay	mean 11.24 days	mean 16.43 days	0.002
(Range)	(3–24 days)	(9–42 days)	
Early complications flap			
No complications	47 (92.2)	17 (81.0)	0.22
Major complications (≥Clavien–Dindo IIIa)	4 (7.8)	4 (19.0)	(Fisher)
Overall early complications			
No complications	45 (88.2)	15 (71.4)	0.082 (χ^2^)
Major complications	6 (11.8)	6 (28.6)	0.318 (Fisher)


Length of stay for abdominoperineal excision


The length of stay was not normally distributed for both direct suturing and VY flap reconstruction, as assessed by the Shapiro–Wilk test, where *p* < 0.001. There were 50 patients in the group without flap reconstruction (n = 50, along with one death) and 21 patients in the group with flap reconstruction (n = 21). The stays were shorter among the first group (mean LOS = 11.24 days) than the second (mean LOS = 16.43 days). A Mann–Whitney U Test was used to evaluate the differences in the lengths of stay between the two groups, and they were a statistically significant, where *p* = 0.002.

### 3.6. Postoperative Complications


Postoperative complications


Early postoperative complications could be distinguished between complications of the surgical site and general postoperative complications. We compared the complications of the surgical site and the overall complications in the group with flap reconstruction and the group without flap reconstruction.

Complications of the perineal surgical site

Fisher’s exact test for association was conducted between the major or severe complications of the surgical site (≥ Clavien–Dindo IIIa) and the reconstruction of the pelvic floor. The results suggested a non-significant connection between the severe complications and reconstruction, where *p* = 0.22. In the group without flap reconstruction, 47 patients (92.2%) had no major complications of the surgical site, while four (7.8%) had major complications (≥ Clavien–Dindo IIIa). In the latter four cases, surgical site infections occurred that required surgical intervention, two of which were not conducted under general anesthesia (a classification of IIIa according to the Clavien–Dindo system) while the other two were (IIIb). In the group with VY flap reconstruction, 17 patients (81.0%) had no major complications of the surgical site and four (19.0%) did experience complications (≥ Clavien–Dindo IIIa). In these four cases, surgical site infections occurred, necessitating surgical intervention. All these interventions were conducted under general anesthesia (IIIb).


Overall early complications


A chi-squared test for association was conducted between the major or severe complications (≥ Clavien–Dindo IIIa) and the reconstruction of the pelvic floor. All the expected cell frequencies were greater than five. The results indicated a non-significant connection between the severe complications and reconstruction, where *p* = 0.082 (Fisher’s exact test obtained *p* = 0.095). In the group without flap reconstruction, 45 patients (88.2%) had no major overall postoperative complications and six (11.8%) did experience complications (≥ Clavien–Dindo IIIa, including surgical site infections). According to the Clavien–Dindo classification system, we found two grade-IIIa cases, three grade-IIIb, and one grade-V case of complications (the last of these resulted in the death of the patient). In the group with VY flap reconstruction, 15 patients (71.4%) had no major complications of the surgical site and six patients (28.6%) did experience complications (≥ Clavien–Dindo IIIa). Based on the Clavien–Dindo system, we identified one grade-IIIa, four grade-IIIb, and one grade-IVb (multi-organ dysfunction) case.

## 4. Discussion

To our knowledge, this is the second largest series describing fasciocutaneous IGAP-based VY flaps for perineal reconstruction in patients that underwent pelvic surgery for malignancy. Our proposal to eliminate extensive dissection of the IGAP perforators simplified the preparation compared to the previous techniques [7,15]. Our results underscored that this could be beneficial for larger defects after ELAPE without compromising the wound healing quality and donor site morbidity. Perineal wound complications such as surgical side infections (SSIs), hernias, or wound dehiscence, as well as persistent perineal sinus following ELAPE for low rectal tumors, remain important causes of increased morbidity and prolonged hospitalization, especially after chemoradiotherapy [15].

Numerous perineal closure concepts and regional flaps following perineal tumor surgery have been described. Vertical rectus abdominis muscles (VRAM), gracilis muscles with or without skin islands (TUG), gluteus muscle flaps and, more recently, perforator flaps such as the deep inferior epigastric perforator flap (DIEP), profunda artery flap (PAP), and superior or inferior gluteal flap (SGAP or IGAP), have been increasingly described in the literature [6,7,8,9,15,16,17,19,20,21,22,23]. The aim of all the flaps is to fill in the pelvic dead space, preventing perineal herniation and bringing well-vascularized tissue into a recipient area almost always altered by pre-irradiation. There has been significant evidence for VRAM flaps, followed by gracilis muscle flaps (TUG) and gluteus maximus muscle flaps—all muscle or myocutaneous flaps [10,22]. There is evidence that a primary closure after ELAPE results in a clinically and statistically higher perineal complication rate than a closure with flaps such as myocutaneous VRAMs or TUGs [9,10,16,21]. In a 2015 a meta-analysis of 10 studies (eight of them used VRAM, while two used gracilis flaps), primary closures (n = 340) were more than twice as likely to be associated with perineal wound complications than plastic reconstruction using myocutaneous flap closures (n = 226; OR = 2.17, 95% CI = 1.34–3.14, *p* = 0.001) [20]. The rate of major perineal wound complications was significantly higher with primary perineal closures (OR = 3.64, 95% CI = 1.43–7.79, *p* = 0.005).

Another systematic review from 2020 evaluated 1827 VRAM flaps for perineal reconstruction, examining 636 VRAM flaps (34.8%) following abdominal perineal excisions of the rectum (34.8%) and the clinical outcomes [10]. A total of 491 patients (26.9%) had complications related to the VRAM flap. No additional subgroup analysis on the individual entities was reported. The mean perineal flap morbidity in the studies was 27% (range = 0–86%), with a complete flap loss rate of 1.8% (n = 33). The most common complication was flap dehiscence (n = 183; 10%). Complications related to the abdominal donor site occurred in a total of 324 (17.7%) patients [10]. Abdominal dehiscence was the most common complication at 5.5% (n = 100) in this review. Other recent study groups reported a 9–17% prevalence of donor site hernias in VRAM patients [13,24]. The donor site morbidity of VRAM flaps and the related unavailability of a laparoscopic abdominal approach led some authors to promote alternative flaps for perineal reconstruction, although it remains a reliable option in the armamentarium of the plastic surgeon. In their 2007 work, Holm et al. recommended a defect closure with myocutaneous gluteal rotation flaps following an extended abdominoperineal resection without compromising the function and innervation of the main muscle portion, which was left in situ [24]. They found that four of the 28 patients had flap-related complications (14.3%) and three had wound infections (10.7%). In general, few studies have compared one flap with another in this setting. A few studies compared gracilis with VRAMs or fasciocutaneous flaps with one systematic review [8,11,25]. In the latter, the success rate of VRAM flaps was higher than that of gracilis flaps (84% versus 64%) [25]. In contrast, Stein et al. showed that gracilis muscle flaps (n = 27) exhibited an equal surgical outcome to VRAM flaps (n = 67) in terms of minor and major complications, mortality, and the complete wound healing time for the reconstruction of pelvic defects [11]. A reported fasciocutaneous gluteal fold flap based on the pudendal artery system seems to cause no more significant complications than a traditional gracilis myocutaneous flap, underlining the benefits of perforator flap concepts with their naturally reduced donor site morbidity over the flaps involving muscle in perineal reconstruction [8]. Inferior gluteal flaps, such as myocutaneous flaps (IGAMs) for perineal closure, were already described by Baird et al. in 1990, one year after Koshima et al. described the first ever performed perforator flap [26,27]. In the meantime, a few studies suggested perforator-based IGAP flaps as reliable options for perineal wound closure [7,15]. On the one hand, in 2012, Hainsworth et al. first described IGAP flaps (n = 40) as a suitable option for perineal defect coverage, with a relatively low rate of 10% for flap-related major complications and another 10% for minor wound complications. On the other hand, Pai et al. described their IGAP concept with an overall perineal early minor complication rate of 25.9% and an early major complication rate of 14.8%. The overall wound healing complication rate was higher (40.74%), similar to Hainsworth et al.’s findings [15].

In our current evaluation, four patients (19%) required surgical interventions due to surgical site infection in the flap group. This was slightly higher than in the other two IGAP flap series. All four patients needed thorough debridement and secondary wound closure, and one patient needed a reposition of the flaps. All four patients were discharged with stable wounds. Compared to the reported myocutaneous reconstructions in a meta-analysis of a 10-flap series with total perineal complications rates for the flaps (16.7–64.7%), the clinical outcomes of our series and the other fasciocutaneous IGAP flaps seemed to be better, with reported pooled means for 34.5% of the complications in the muscle flap series [20]. The rate of flap necrosis and donor site complications, such as an abdominal wall hernia (18.8% in the VRAM group), seemed to be higher as well in the reported myocutaneous flaps compared to the gluteal turnover flaps. Using a multivariable analysis, a multicenter study with 25 gluteal turnover flaps revealed uncomplicated wound healing among 68% of patients (17 out of 25 cases) in the flap group versus 64% (124 out of 194) after the primary closure (OR = 2.246, 95% CI = 0.734–6.876, *p* = 0.156) [16]. No major wound complications required surgical re-intervention after the flap closure. At a 12-month follow-up, none of the 18 included patients had chronic perineal sinus, compared to 6% (11 out of 173 patients) after primary closure (*p* = 0.604). The rate of symptomatic perineal hernias at 18 months was nil after the flap closure, compared to 9% after the primary closure (*p* = 0.184).

This study superbly demonstrated the lack of standardization for the terminology and aggravated comparability regarding the flap of choice and its outcome. The original technique Blok et al. used in their series only de-epithelialized 2–3 cm in semilunar medial fashion for one perineal wound border, which was then turned into the cavity without any further flap elevation [17]. From the standpoint of a plastic surgeon, this cannot be compared to standardized plastic surgery flap techniques. Furthermore, to our understanding, the term ‘turnover’ is also used incorrectly [17]. In real turnover flaps, such as the pectoralis turnover flap for sternal wound coverage, the lateral part of its muscle region is turned over the medial part of the flap to fill out a cavity in the midline. With Blok et al.’s technique, a deep plumbing of the pelvic cavity and the closure of the wide defects without tension is not possible.

In general, the comparison of the different studies with flap coverages in perineal defect situations is often complicated by heterogeneous patient populations and methodological differences, as shown above and discussed by the authors of each cited systematic review or meta-analysis [10,20,25,28]. The methodological issues mainly included the retrospective data available, degree of radiotherapy, and extent of resections, and in turn, a high risk for selection bias. For this reason, among others, there was no consensus to date on which flap is favored for defect coverage following rectal extirpation. This issue is also well addressed and summarized in a consensus paper by the Association of Coloproctology of Great Britain and Ireland (ACPGBI) [28].

To our understanding, the IGAP flap concept retains the pros of favorable clinical outcomes for flap coverages in the perineal region while minimizing the typical donor site morbidities associated with myocutaneous flaps, all without compromising the overall clinical outcome.

In our approach, the decision on whether a flap was needed was part of the surgery plan, meaning that a plastic surgeon was always on standby. The final decision was made during the tumor resection by the general surgeon if no sufficient or tension-free wound closure could be reached or in cases of wide cavities with high risk for fluid retention, SSIs, and pelvic hernias.

The limitations of this study were its inevitable retrospective character and its inability to randomize myocutaneous flaps with IGAP flaps. Furthermore, the selection bias in deciding whether a flap was needed was subject to a single surgeon’s opinion. A possible bias of the results due to this, and to the fact that the surgeries in the study were not all conducted by the same surgeons, cannot be excluded.

## 5. Conclusions

The IGAP flap appears to be a good alternative to VRAM flaps, given its low donor morbidity and documented surgical advantages. In comparison to patients with non-plastic closures, the oncological outcomes seem to be better in patients who have received plastic reconstruction. Therefore, as the need for plastic coverage is indicated by the rectal surgeon, every center should have access to surgeons with knowledge of plastic reconstruction in an interdisciplinary approach. Some difficulties include the lack of standardization possibilities and the surgeon’s personal decisions on the type of adopted defect coverage.

## Figures and Tables

**Figure 1 cancers-15-03345-f001:**
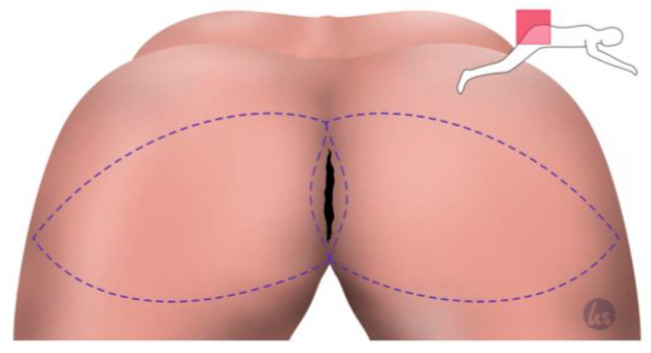
Planning of the flap in the VY manner. The flap size should always be of an appropriate size to cover and line the wound cavity well. The caudal incision should be created within the inferior gluteal fold. The tip of the V flap is placed over the trochanter to gain sufficient tissue for plication.

**Figure 2 cancers-15-03345-f002:**
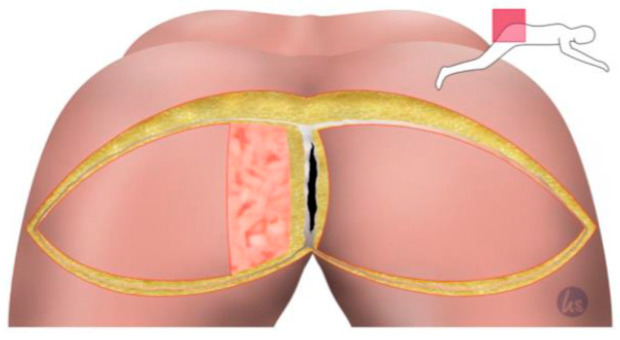
Sharp skin incision and dissection down to the gluteal fascia and incision of the gluteal fascia. The muscle was left untouched. Then, one flap side can be easily transposed into the deep cavity. The de-epithelialization of the necessary area medially after manual transposition, measuring and marking the area to be plated into the deep cavity.

**Figure 3 cancers-15-03345-f003:**
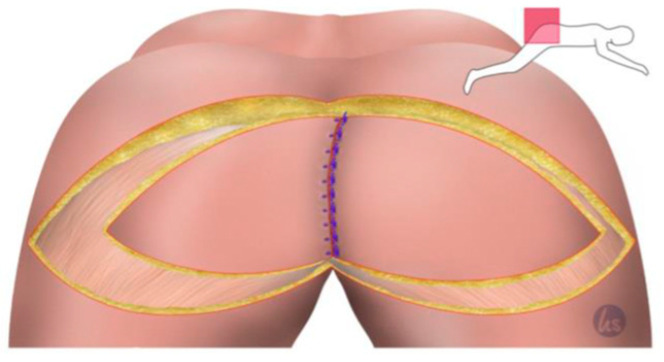
The de-epithelialized flap can now be sutured in depth. For this, we used a 0 non-absorbable suture. The VY flap from the other side was used to cover the defect from the contralateral side and in the parts of the de-epithelialized area of the other flap in the midline. The midline was closed in three rows, 0 s absorbable sutures deep, 2-0 absorbable for the subcutaneous closure, and 0 s non-absorbable sutures for the skin closure. This was important to adequately accommodate the tension in the midline.

**Figure 4 cancers-15-03345-f004:**
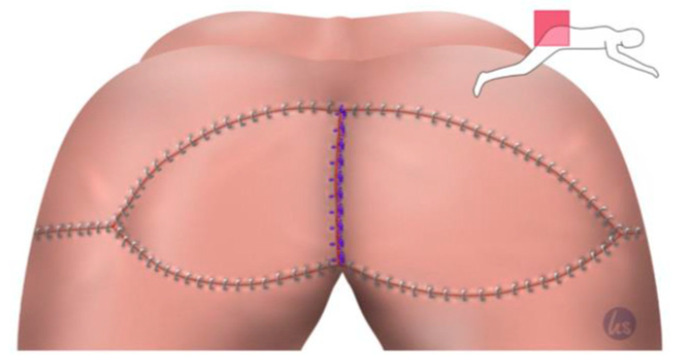
Closure of the remaining Y-flap portions also in three rows. The 0 s absorbable suture addressed Scarpa's facia. A 2-0 absorbable suture for the subcutaneous closure was used and skin staples for the skin suture using an intracutaneous 3-0 absorbable monofilament suture.

## Data Availability

The data can be shared up on request.

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
