# Peer review of "Feasibility of Perineal Defect Reconstruction with Simplified Fasciocutaneous Inferior Gluteal Artery Perforator (IGAP) Flaps after Tumor Resection of the Lower Rectum: Incidence and Outcome in an Interdisciplinary Approach"

_cancers, 2023, doi:10.3390/cancers15133345_

Round 1
Reviewer 1 Report
This is a single centre retrospective cohort study analyzing 101 patients who underwent ELAPE during a 4 year period, with proportion perineal reconstruction and hospital stay as main outcomes. 72 patients underwent primary closure, and 29 IGAP flap in VY-fashion. The authors report favourable outcomes after IGAP flap reconstruction, and conclude that this is a reliable, technical easy and safe option for perineal closure. This conclusion is justified based on the presented data, although I do not agree with the authors if they state that the complication rate was "acceptable", given the fact that 9/27 patients had a Clavien Dindo III complication.
There are increasing numbers of such small cohort series reporting on a certain flap closure technique in patients undergoing APE. However, this study is still of additional value since the data on fasciocutaneous VY gluteal flap reconstruction is relatively limited. Others have described musculocutaneous gluetal flaps, but this technique is interesting because of the low donor site morbidity. Furthermore, it can be combined with a laparoscopic abdominal approach which makes it a good alternative for the much more invasive rectus abdominis flaps.
Some detailed comments:
- the abstract can be more structured. The results are not separately reported for primary closure and IGAP reconstruction groups. Furthermore the flap related complication rate is missing.
- the introduction consists of just one single paragraph, but might also benefit from a better structure with separate paragraphs to improve its readability.
- the technical description of the flap is now sometimes formulated as a narrative review with recommendations. Please rewrite, by just stating what was performed in the patient series.
- the statistical paragraph is just one sentence. This should also include statements on descriptive statistics and which tests and p-value cut-off were used for comparative analyses.
- "mittelwert" should be translated (results section)
- the complications paragraph needs better structure. Do not start with a qualitative statement (... was low). It is stated twice that there were 2 IIIa and 7 IIIb complications.
- if reporting on perineal hernia, the median follow-up and interquartile range should be reported as well. Furthermore, for the two patients with perineal hernia, the authors can provide the interval from index surgery and who this was treated.
- Table 1 should have two columns with the baseline data separately displayed for the primary closure and the IGAP flap reconstruction groups, with p-values of appropriate statistical tests for comparisons.
- It would be informative to have a second table with clear data on outcomes (e.g. duration of surgery, blood loss, length of stay, complications, reinterventions) after both primary closure and flap closure, also with statistical comparison between the two groups.
- There is an extensive description of literature in the discussion, but the authors should better discuss their own results using this literature as a reference. Why did they find a relatively high complication rate?
Author Response
Dear Reviewer,
thank you for your comments. We changed all issues. The paper is now much better to understand:
the abstract can be more structured. The results are not separately reported for primary closure and IGAP reconstruction groups. Furthermore the flap related complication rate is missing.
- The abstract is now better structured and the complication rate is shown
- the introduction consists of just one single paragraph, but might also benefit from a better structure with separate paragraphs to improve its readability.
- - We changed tho whole introduction for a better understanding
- the technical description of the flap is now sometimes formulated as a narrative review with recommendations. Please rewrite, by just stating what was performed in the patient series.
- - we changed this passage completely
- the statistical paragraph is just one sentence. This should also include statements on descriptive statistics and which tests and p-value cut-off were used for comparative analyses.
- - This passage was completely changed
- "mittelwert" should be translated (results section)
- the complications paragraph needs better structure. Do not start with a qualitative statement (... was low). It is stated twice that there were 2 IIIa and 7 IIIb complications.
- -we changed this passage completely
- if reporting on perineal hernia, the median follow-up and interquartile range should be reported as well. Furthermore, for the two patients with perineal hernia, the authors can provide the interval from index surgery and who this was treated.
- This part is part of our ongoing study and will be published soon. It is not part of the actual study.
- Table 1 should have two columns with the baseline data separately displayed for the primary closure and the IGAP flap reconstruction groups, with p-values of appropriate statistical tests for comparisons.
- - we changed the Tables completely and added 3 more
- It would be informative to have a second table with clear data on outcomes (e.g. duration of surgery, blood loss, length of stay, complications, reinterventions) after both primary closure and flap closure, also with statistical comparison between the two groups.
- - we did so. It is much more informative
- There is an extensive description of literature in the discussion, but the authors should better discuss their own results using this literature as a reference. Why did they find a relatively high complication rate?
- We discussed our complication rate in the „discussion“ section related to the other outcomes. Some of the patients with VY-reconstruction were exenteration operations with extensive local surgery
- I added more references to the introduction and structured it new
- The whole manuscript was professional proofreaded by an native speaker
Reviewer 2 Report
This retrospective cohort study describes the outcome of IGAP in patients undergoing eAPR for malignant and benign diseases. First of all, I would like to commend the authors for their work. However, I do have some concerns.
Major
- The resection of a malignant rectal cancer tumor greatly differs from that of a melanoma or anal cancer resection. As with rectal cancer, most of the ischioanal space is being preserved, which is mostly resected in anal cancer. This will lead to a difference in perineal wound morbidity. In addition, neoadjuvant chemotherapy greatly differs among different tumors and has a substantial effect on the outcome
- what does this add to the existent literature?
- The results section is too limited, and confounding factors should have been taken into consideration. This means that subgroup analysis should have been performed.
- how long was the follow-up
-"The statistical section of the manuscript states 'Statistical analysis was done using SPSS." The statement in the manuscript about the use of SPSS for statistical analysis is not presented in the standard format for reporting statistical analysis in scientific articles. It implies that the manuscript may not have followed the typical conventions for presenting statistical analysis, which could make it more difficult for other researchers to understand or reproduce the study's result
- the results in the table does not always add up to 101
Minor
- Results of the study should not be mentioned in the method section (i.e number of included patients)
- references should be added to the introduction
- there are a lot of grammatical errors (i.e sentence 55 the and The; Aim of the procedure is to move the area of resection as is incorrect; oncological background sentence 151 result section)
- a R1 percentage of 9.8% is worrisome
- paragraph 3.3 "complications after plastic reconstruction was low" what is low? how much?
- please provide “denominator" and "numerator”
-
The manuscript contains several errors in grammar, spelling, and punctuation. It is poorly organized and difficult to follow. The writing quality should be improved to ensure that the study's methods, results, and conclusions are clearly and effectively communicated.
Author Response
Major
- The resection of a malignant rectal cancer tumor greatly differs from that of a melanoma or anal cancer resection. As with rectal cancer, most of the ischioanal space is being preserved, which is mostly resected in anal cancer. This will lead to a difference in perineal wound morbidity. In addition, neoadjuvant chemotherapy greatly differs among different tumors and has a substantial effect on the outcome
- what does this add to the existent literature?
- This technique allows a very good coverage of the resected space and the combination with minimally invasive pelvic surgery is one the best opportunities. There are just a few reports. This study adds a lot of information.
- The results section is too limited, and confounding factors should have been taken into consideration. This means that subgroup analysis should have been performed.
- The result section is completely revised. 4 Tables show the patient data and the postoperative outcome.
- how long was the follow-up
- - we added this information to the result section.
-"The statistical section of the manuscript states 'Statistical analysis was done using SPSS." The statement in the manuscript about the use of SPSS for statistical analysis is not presented in the standard format for reporting statistical analysis in scientific articles. It implies that the manuscript may not have followed the typical conventions for presenting statistical analysis, which could make it more difficult for other researchers to understand or reproduce the study's result.
-The statistical analyses were completely revised
- the results in the table does not always add up to 101
- We revised this section completely
Minor
- Results of the study should not be mentioned in the method section (i.e number of included patients)
- We changed this information
- references should be added to the introduction
- we added references to the introduction
- there are a lot of grammatical errors (i.e sentence 55 the and The; Aim of the procedure is to move the area of resection as is incorrect; oncological background sentence 151 result section)
- a R1 percentage of 9.8% is worrisome
We added a table in the result section. Marginal resection is sometimes the only choice in extensiv disease and need of exenteration. In normal rectal excision this is lower as shown in the results.
- paragraph 3.3 "complications after plastic reconstruction was low" what is low? how much?
- please provide “denominator" and "numerator”
- -->We added a new table
Reviewer 3 Report
The study is well done, the material is large enough and the methods look reliable. However the study is based on extensive and very recent literature, gives some new information and this warrants its publication.
Author Response
Thank you so much. We added additional information for even better understanding